# A Neural Network Computational Spectrometer Trained by a Small Dataset with High-Correlation Optical Filters

**DOI:** 10.3390/s24051553

**Published:** 2024-02-28

**Authors:** Haojie Liao, Lin Yang, Yuanhao Zheng, Yansong Wang

**Affiliations:** 1Institute of Frontier and Interdisciplinary Science, Shandong University, Qingdao 266237, China; 202100120226@mail.sdu.edu.cn (H.L.); 202221101@mail.sdu.edu.cn (Y.Z.); wangyansong@sdu.edu.cn (Y.W.); 2Institute of Space Sciences, Shandong University, Weihai 264209, China

**Keywords:** computational spectrometer, neural network, small training dataset, spectra reconstruction

## Abstract

A computational spectrometer is a novel form of spectrometer powerful for portable in situ applications. In the encoding part of the computational spectrometer, filters with highly non-correlated properties are requisite for compressed sensing, which poses severe challenges for optical design and fabrication. In the reconstruction part of the computational spectrometer, conventional iterative reconstruction algorithms are featured with limited efficiency and accuracy, which hinders their application for real-time in situ measurements. This study proposes a neural network computational spectrometer trained by a small dataset with high-correlation optical filters. We aim to change the paradigm by which the accuracy of neural network computational spectrometers depends heavily on the amount of training data and the non-correlation property of optical filters. First, we propose a presumption about a distribution law for the common large training dataset, in which a unique widespread distribution law is shown when calculating the spectrum correlation. Based on that, we extract the original dataset according to the distribution probability and form a small training dataset. Then a fully connected neural network architecture is constructed to perform the reconstruction. After that, a group of thin film filters are introduced to work as the encoding layer. Then the neural network is trained by a small dataset under high-correlation filters and applied in simulation. Finally, the experiment is carried out and the result indicates that the neural network enabled by a small training dataset has performed very well with the thin film filters. This study may provide a reference for computational spectrometers based on high-correlation optical filters.

## 1. Introduction

Conventional spectrometers are bulky due to complex optical paths and moving parts that hinder such spectrometers from achieving wider applications in handheld, spaceborne, and airborne scenarios. Contrarily, the proposed computational spectroscopy conceptualization, based on the principle of compressed sensing (CS), has significant potential for portable applications, such as remote sensing, healthcare, and astronomical observation [1,2]. These applications are in high demand because of their high spectral resolution, compactness, and low cost. The computational spectrometer can break the confinement from hardware technology and improve efficiency and accuracy, based on its effective design technique and efficient algorithms. In the field of computational spectroscopy, an unknown signal is measured according to CS theory by projecting the unknown signal onto a random basis. The filter structure, which acts as the encoding part, has a specific response function. The transmission spectrum has a randomly distributed spectral signature due to multiple reflections at the interface of thin film filters or nanostructures [3]. Each different configuration of the filter structure produces a unique response function, which results in a completely different response function that is used on a random basis [4].

The major problem is the technical complexity of optical filters with random transmittance spectra on the broadband spectrum, and the second problem is computation efficiency. The performance of CS relies on the randomness of the measuring bases, namely the encoding part. The transmittance of the optical filters of the computational spectrometer should have diverse spectral features over the broadband spectrum range. Thus, we are looking for a sizable group of filters with minimal correlation. The low correlation or non-correlation of the transmittance between any two of the spectrometer pixels is basically the prerequisite for the application of CS [5,6]. This brings a challenge to the design and fabrication of thin film filters and nanostructures like photonic crystal slabs, metasurfaces, and thin film filters [7]. This problem also leads to poor performance of conventional iterative reconstruction algorithms such as gradient projection for sparse reconstruction (GPSR), orthogonal matching pursuit (OMP), and subspace pursuit (SP) algorithm [8,9,10]. Neural networks (NNs) have been introduced because of their various advantages. Bao [11] combined conventional algorithms with an NN, named solver-informed NN, to achieve better alignment of the reconstructed spectral. Even though the bulky volume of the large training dataset hinders its widespread application. Many researchers have made some progress with NNs in computational spectroscopy. Zhang [12] proposed a broadband encoding stochastic camera featuring fully connected NN layers. Ding [13] proposed an encoding and reconstruction convolutional NN (CNN), named wide-spectrum encoding, and a reconstruction NN named wide-spectrum encoding and reconstruction neural network (WER-Net). Both Zhang and Ding’s networks are trained by approximately 1,650,000 spectral datasets from the Columbia Imaging and Vision Laboratory (CAVE) [14] and the Interdisciplinary Computational Vision Laboratory (ICVL) [15]. Kulkarni applied an NN in the field of image compression reconstruction [16], which was the first study that solved the CS problem with NN. Subsequently, Song introduced deep-learned broadband encoding stochastic filters for computational spectroscopic instruments [12], the NN is also trained by the heavy training dataset CAVE and ICVL. Bao used a NN to improve the reconstruction accuracy of the conventional spectral reconstruction algorithm by fitting the original spectral curve and the reconstructed curve obtained from the conventional iterative algorithm [11]. The loss functions for almost all the NNs used in the computational spectrometer are all about the mean squared error (MSE) between the original curves and the reconstructed spectral curves. It enlightens us that there may be a possibility of getting rid of the compulsory demand for non-correlation properties in computational spectroscopy. To address the aforementioned problems, this study introduces an NN computational spectrometer trained by a small dataset with high-correlation optical filters.

The remainder paper is organized as follows: Section 2 presents the NN architecture and methodology. Section 3 describes the training and simulation. Section 4 reports the experimental results. Finally, Section 5 concludes this study.

## 2. Theoretical Model and Design Methodology

For traditional computational spectrometers, iterative algorithms based on the CS theory have been developed for many years. The encoding procedure is completed by random optical filters and focal plane detectors which are alike in various spectrometers. In this way, satisfying the incoherence criterion requires minimum correlation between any two of the spectrometer filters, which brings great challenges to the design and fabrication of broadband filters. Contrarily, the training process of the filters in NNs only pays attention to the conformity of the reconstructed spectrum and the ground truth, which does not require the incoherence criterion to be satisfied. The NNs proposed in this study can achieve very good reconstruction accuracy with highly correlated broadband optical filters. The decoding procedure, which is completed by the reconstruction algorithm of an NN, often requires storage in megabytes, and the reconstruction procedure takes up computing resources. So the architecture needs to be as simple as it can be.

Neurons comprise the structure of a unit of artificial NNs, based on the unit of deep learning to simulate the working process of biological NNs. The McCulloch–Pitts (M–P) neuron model, proposed by McCulloch and Pitts [17], is the most commonly used to develop the computational model of neurons. Hopfield used NNs to solve NP-hard problems for the first time [18], helping to achieve rapid development in NNs. LeCun proposed LeNet-5, the standard CNN, which greatly contributed to the development of NNs [19]. Since then, deep learning has started to spring up in various fields. Among them, Kulkarni applied deep learning to the field of image compression and reconstruction for solving CS problems [16]. Zhang applied feedforward NNs with all fully connected layers to the field of spectral reconstruction [12]. Ding proposed a lightweight CNN for computational spectroscopy [13].

Since the proposed neural network featured with trained by a small training dataset, we then refer to it as STD-Net. The architecture of the STD-Net is presented as follows: The first layer, which serves as an encoding layer, is a matrix of known transmittance curves for 15 polymethyl methacrylate (PMMA) filters. The second to the fifth layers are the reconstruction layers (referred to as decoding layers), and each layer is followed by the Leaky ReLU as the activation function. The overall reconstruction network structure is summarized as FC (encoding without bias) − (FC − LEAKY_RELU) × 4, as described in Figure 1.

### 2.1. CNN and All FC Network

The CNN and all FC networks are first compared, based on the analysis and verified by training process and reconstruction performance before focusing on all FC layers in STD-Net. CNN is more suitable for image segmentation because it uses two strategies of local connectivity and weight sharing to reduce the network model complexity. CNN also has the property in which the extracted features have a certain degree of output invariance to change the input data, such as translation, rotation, and scale scaling. In more depth, local connectivity means that each node in a convolutional layer is connected to only part of its predecessor layer and learns only local features of the input data. This is because the correlation between image pixels is related to the distance between pixels, and the correlation is strong between pixels that are closer together. However, in the field of computational spectroscopy, the data input to the decoding layer all represent some feature of the measurement spectrum in each band and are correlated with each other, and learning only partial features is not sufficient to fully utilize the encoded data. In addition, weight sharing refers to the use of the same convolution kernel to convolve different regions of the input matrix to detect the same feature. These two strategies result in localized features of the input matrix that are independent of the position of the data composing the features in the matrix. Thus, when the data in the matrix is moved, the convolutional layer still finds the same feature, only the position is changed.

However, the output invariance of the CNN leads to a loss of reconstruction accuracy in the computed spectrum. In the encoding layer, the original spectral curve is compressed and sampled by 15 filters to obtain 15 data, and their arrangement order is the same as the arrangement order of the filters, the same local data features may represent different spectral curve information because of their different positions in the matrix, and the CNN treats them as the same information due to the output invariance and thus leads to the reduction in reconstruction accuracy. Therefore, we canceled the method of scanning the input matrix by the convolution kernel and changed the size of the convolution kernel to be the same as the input matrix, which is computed in the same way as the fully-connected layer, so we replaced the convolution layer of the reconstruction algorithm by the fully-connected layer.

On the other hand, some studies suggest that convolutional layers can reduce the number of parameters to improve computational efficiency [13,20]. However, the convolutional kernel scanning method increases a large number of operations and computations by multiple convolutions of the input matrix compared to the fully connected layer, which reduces the computational speed and requires a higher performance of the equipment.

### 2.2. Small Training Dataset

A proposed presumption in this study is that the common spectral large training dataset such as CAVE and ICVL has a certain distribution law, approximate or consistent with the distribution law of natural spectra. Therefore, a small training dataset, randomly drawn from a large training dataset of the same proportion, should have similar distribution laws as the entire training dataset. This shows that this small dataset contains the main features of the whole training dataset. To analyze and find this distribution law, we introduce the concept of the Pearson product-difference correlation coefficient, whose mathematical expression is as follows:(1)$$\rho \left(x,y\right)=\frac{\sum_{i=1}^{n}\left(x_{i}-\bar{x}\right)\left(y_{i}-\bar{y}\right)}{\sqrt{\sum_{i=1}^{n}{\left(x_{i}-\bar{x}\right)}^{2}}\sqrt{\sum_{i=1}^{n}{\left(y_{i}-\bar{y}\right)}^{2}}}$$
where x and y each represent a spectral curve; *x_i_* and *y_i_* refer to the intensity corresponding to the *i*-th wavelength of the spectral curve, and $\bar{x}$ and $\bar{y}$ represent the average intensity of the spectral curve.

The Pearson product difference correlation coefficient has a value between −1 and 1. The closer its absolute value to 1, the stronger the linear correlation between variables x and y.

The whole dataset is a hybrid dataset of CAVE and ICVL. CAVE dataset covers the spectra of five scenes: objects, skin and hair, paint, food, and drink, and has been cited by more than 632 literature sources. ICVL dataset covers the spectral information of nature, trees, and buildings, and has been cited by more than 415 pieces of literature. The mixture of the two, with a total of 1.65 million spectral curves, contains spectral information of the most commonly used scenes and is also widely recognized and used by the image processing industry for satisfying experimental requirements. Since the spectral resolution of the dataset is only 10 nm, which is somewhat different from the ideal resolution, we use the least squares fitting method to increase the spectral resolution to 2 nm to improve the resolution of the spectral reconstruction algorithm. Then, based on the assumptions made in the previous section, we take the absolute average of the correlation coefficient between each spectral curve in the dataset and the other spectral curves and are regarded as the distribution weight of that spectral curve in the large training dataset to derive the spectral distribution, and divide the large dataset into several intervals based on their distribution weights, to obtain the small dataset for training, testing, and simulation.

### 2.3. Loss-Function and Training Methods

To assess the conformity level between the reconstructed spectral curve and the ground truth, we adopt MSE as the objective function:(2)$$\Theta =argmin{\left\|S-\hat{S}\right\|}_{2}$$
where *S* denotes the input actual spectral matrix, and $\hat{S}$ represents the output reconstructed spectral matrix.

Since its loss function focuses on the alignment between the input and output spectral curve, the focus here is not on the optical filters’ non-correlation property. Luckily, the non-correlation property of optical filters is only a sufficient condition but not necessary for computational spectroscopy. That explains why the NN computational spectrometer can achieve good performance later even with high-correlation optical filters in this study.

A high learning rate is capable of accelerating learning in the early stage of algorithm optimization, facilitating the model to approach the local or global optimal solution for the loss function. However, this approach can cause the model to fluctuate too much in the later stage and prevent the model from converging to the optimal value. The learning rate adaptive decay mechanism throughout the training process is applied, which takes the average reconstruction accuracy MSE of each training epoch as the index and reduces the learning rate when the MSE does not decrease to a noticeable extent. This mechanism can effectively reduce the model fluctuations in the middle and later stages of training, making the model closer to the optimal solution.

In addition, the batch gradient descent model uses all samples to update the gradient in each iteration, resulting in a stable model iteration. However, the iteration speed is very slow for a sizable training dataset. In contrast, the stochastic gradient descent method iterates once after calculating the loss function for one training data, which has a fast iteration speed at each step, but with poor stability. We used the small-batch stochastic gradient descent method to coordinate the model stability and training speed. Thus, we achieved strong stability of the optimization direction while ensuring the training speed after summing up the loss function and iterating the gradient immediately.

## 3. Training and Simulation 

### 3.1. Small Training Dataset Establishment

The CAVE and ICVL datasets mentioned in Section 2.2 have a spectral profile resolution of only 10 nm, which falls short of the experimental accuracy requirements. Therefore, we increase the spectral resolution to 2 nm by average interpolation based on extracting the spectral curve data in the visible band, i.e., between 400 and 700 nm. Based on the assumptions in Section 2.2, the whole dataset with 1.65 million spectral curves is regarded as a large training dataset, and the correlation coefficients between each spectral curve and the other spectral curves are taken as the absolute average value and are considered as the distribution weight of that spectral curve in the large training dataset to derive the contribution of that spectral curve to the richness of the whole dataset. The lower the distribution weight, the higher the contribution. The large training dataset is divided into 1000 intervals in order of the distribution weights of each spectral data in fixed steps. The overall distribution is shown in Figure 2.

We extract a certain number of spectral data to establish the small training dataset, according to the distribution proportion in each interval. The rest of the large training dataset is delegated as a testing dataset and experimental dataset. The training dataset serves as the small training dataset for NN model training. The testing dataset is used for the model to observe the current model accuracy and to judge whether the model is overfitting after each epoch of the training process. The experimental dataset is used for the reconstruction of the experimental accuracy after the NN parameters are fixed. According to the different proportions of the training and testing datasets, four groups of experiments were conducted, as shown in Table 1.

The training dataset is extracted by 90%, 10%, 5%, and 3% of the entire dataset, based on the proportion of each interval, using 1,450,000, 164,540, 82,021, and 49,007 spectral curves. Most of the remaining datasets of each interval are selected as the testing dataset, accounting for 89%, 94%, 96%, and 9% of the intervals, with 1,469,467, 1,551,986, 1,585,000, and 149,007 spectral curves, respectively. The experimental datasets of the four groups are the same; all of which are selected from the 1% of each interval of the entire dataset randomly, with 15,993 spectral curves.

### 3.2. Encoding Layer

Figure 3 shows the 15 selected optical filter transmittance curves, and Table 2 is the correlation coefficient matrix. The larger the correlation coefficient, the deeper the color in Table 2. This process shows that the optical filter transmittance curves are highly smooth, with significant variation and high richness. Although the maximum value of the correlation coefficient in Table 2 reaches 0.7, the MSE of reconstruction accuracy in the experimental results is still high. This is noteworthy since NN helps break the constraint of the intense demand for high non-correlation between any two of the optical filter transmittance curves. This process surely facilitates the design and fabrication of optical filters in the field of computational spectra.

### 3.3. Training and Simulation

In the training process, we take the average reconstruction accuracy MSE of each training epoch as the index and multiply the learning rate by 0.5 when the MSE does not decrease in 2 consecutive epochs. The model gets to the optimal solution very soon. In addition, the small-batch stochastic gradient descent method is applied to randomly and non-repeatedly select 64 data from all training data to coordinate the model stability and training speed.

The training process of the decoding network lasted until the 200th epoch. The MSE of training and testing is shown in Figure 4. At the 19th epoch, the training and testing MSE is already below 1 × 10^5^. In the middle of training, the downward trend of the loss function of the test has been reduced to close to 0. Subsequent training is less helpful in reducing the loss function of the test, which can be assumed that the test error at this time is the same as the training error at the end of training. At this time, the MSE of the test reaches 3 × 10^6^, and the comparison in Table 3 can be found to be a sufficiently high reconstruction accuracy.

Simulated spectra are reconstructed with NNs trained on each of the four datasets to investigate the effect of the size of the training dataset on the spectral reconstruction accuracy. In order to simulate the error in measuring the filter transmittance profile and spectra, we add random Gaussian noise with mean 0 and standard deviation (σ) of 10^−3^ and 10^−2^ to the encoding network, namely the filter transmittance profile matrix. Afterward, we compare them with the NN (STD-Net) without being doped with Gaussian noise to derive the average MSE, full width at half maximum (FWHM), peak amplitude error(PAE), peak wavelength position deviation(PWPD), and reconstruction speed of the reconstructed spectra versus the spectra in the simulated dataset, and record the data in Table 3. Some important conclusions are stated as follows.
(1)The larger the dataset, the better the performance in metrics such as MSE, FWHM, PAE, and PWPD, and the better resistance against noise interference.(2)Small datasets also show good accuracy and have shorter training processes under the condition of satisfying certain accuracy.(3)The reconstruction speed is approximately identical when the network architectures are the same, and no correlation exists with the size of the training dataset.

These enlighten us that in some new fields where it is difficult to build large datasets or where the training cost is extremely high, the rational use of small datasets is important for cost saving.

## 4. Experiment and Discussion

### 4.1. Experiment Setup and Results

The prototype of the experimental system for the computational spectrometer is shown in Figure 5. It is composed of the light source, the camera module, the lens, the filters, and the color card. In the experimental system, the standard color card is used as the sample, the camera model is MER-051-120U3M/C (DAHENG Image, Beijing, China) with a resolution of 808(H) × 608(V) and an image element size of 4.8 μm × 4.8 μm, while the lens model is C02812F16-3MP (ANNAI Technology, Shenzhen, China) with an aperture of 1.6 and a focal length of 2.8 mm. The PMMA filters with known transmittance are placed in front of the detector lens to acquire gray-scale images. During every imaging, the filters are switched manually. The obtained measurement data is then entered into the computer for reconstruction. Performance metrics include average MSE, FWHM, peak amplitude error, peak wavelength position deviation, and reconstruction speed.

The data in Table 4 shows some metrics to display the accuracy and speed of spectral reconstruction for each group in this experiment:(1)The accuracy metrics in Table 4 are about 10% worse than the simulation results in Section 3.3, but the spectral curves essentially overlap as seen in Figure 6. Results of spectral curve reconstruction, and the loss of accuracy are within acceptable limits.(2)The training dataset for the group-large is 9 times larger than the group-α. However, their MSEs are of the same order of magnitude, reaching an acceptable range. This proves the correctness of the small training dataset assumption, reveals that it can learn the main features of the entire training dataset, provides a reliable dataset minimization method, and reduces the training cost of neural networks.(3)The reconstruction speed is approximately identical when the network architectures are the same, and no correlation exists with the size of the training dataset.

### 4.2. Comparison with Other Algorithms

We chose the classical GPSR and the Orthogonal Matching Pursuit (OMP) algorithm for comparison. As for the measurement, we introduced the Gaussian random matrix and the known encoding network. The reconstruction is performed on the Intel i5-12490f platform. For NN algorithms comparison, we chose the parameter constrained spectral encoder and decoder (PCSED) [12] and WER-Net, whose model parameters were trained using CAVE and ICVL. They work on the NVIDIA RTX3080 platform, as does STD-Net. The spectral profiles for the reconstruction experiments were obtained from the results of the spectrometer shots of the natural environment, captured by the ISPECFIELD-HH spectrometer. The reconstruction results are provided in Table 5.

In the GPSR and OMP algorithms, the average MSE reaches the order of 1 × 10^−2^ and 1 × 10^−3^, respectively, based on ideal Gaussian random matrices. In contrast, the reconstruction accuracy of STD-Net (group-β) reaches the order of 1 × 10^−6^, which is 1863 times higher than that of the GPSR and 723 times higher than that of the OMP. In terms of computing time, the single-spectrum reconstruction time of STD-Net (group-β) is 1.79% of that of the GPSR and 13.3% of that of the OMP. This proves that STD-Net substantially outperforms the conventional GPSR and OMP algorithms in terms of reconstruction accuracy and reconstruction time. Furthermore, the compression matrix used in the GPSR and OMP is a random Gaussian matrix. With the current industrial technology, it is not possible to produce and manufacture filters with completely randomly distributed transmittance curves. However, the transmittance curves simulated by STD-Net are feasible for industrial production.

The PCSED and WER-Net reconstruction accuracy is acceptable because the MSE of their reconstruction accuracy reaches 1 × 10^−4^ and 1 × 10^−5^, respectively. Moreover, the reconstruction accuracy of STD-Net is better as it reaches 10^−6^. The filter transmittance curves, obtained through WER-Net training, show significant abrupt changes and lack smoothness, making it problematic in industrial production. In Section 2.1, we conclude that the characteristics of the CNN harm the reconstruction accuracy because of its incompatibility with the spectral compression reconstruction task. Compared with FC networks, CNN occupies a substantial amount of video memory and requires high hardware performance owing to a large number of arithmetic operations. Taking WER-Net as an example, for a single spectral curve input, the computation of the three convolutional layers has reached 1.9 Mflops. After replacing them with a fully connected network, the computation of the whole decoding network only reaches 0.216 Mflops, which reduces the computation by 88.8%. In terms of computing time consumption, the single-spectrum reconstruction time of STD-Net (group-β) is 42.4% of that of WER-Net, and 10.7% of that of PCSED. Thus, the MSE of the reconstruction accuracy is improved to 10.17 times that of WER-Net and 110.67 times that of PCSED. The experimental data demonstrates that STD-Net substantially improves the speed and accuracy of spectral reconstruction.

Regarding the proposed STD-Net, even group-γ, which has the worst performance, its MSE still reaches 9 × 10^−6^ with only 3% of the entire dataset, which is sufficiently superior in terms of network training time and data demand, demonstrating the advantages of a small training dataset.

## 5. Conclusions

This study proposes an NN computational spectrometer with high-correlated optical filters. It consists of high-correlation optical filters for encoding and a neural network called STD-Net for decoding, which is trained on a small training dataset. First, we propose a presumption that the spectral has a specific distribution law, in which a small training dataset, composed of randomly extracted data from the entire dataset, can cover the main features. Based on this, targeting the CAVE and the ICVL spectral datasets, the correlation coefficients of each spectral curve, and all other spectral curves are averaged and processed in absolute value as the distribution weight. All data are divided into 1000 intervals. Then, 90%, 10%, 5%, and 3% of the data are randomly extracted for training while certain portions of the dataset work as the test dataset in case of overfitting in the training process. In addition, a tailored loss function and adaptive learning rate mechanism are introduced to improve the training efficiency and reconstruction accuracy. Since this part, only focuses on the MSE between the original curves and the reconstructed ones. It has no compulsory request on the non-correlation property of optical filters. The non-correlation property for optical filters is a sufficient but unnecessary condition. Then the highly correlated PMMA filters are applied to work as the encoding layer. In the reconstruction network, we proposed a four-layer FC architecture network. Multi-scale training datasets are established to train the constructed neural network. Furthermore, we conducted four groups of comparison experiments with different data extraction ratios, indicating that STD-Net could achieve high reconstruction accuracy and robustness, even when the training data quantity was immensely limited. Finally, an experimental system was introduced, and the results indicate that the proposed NN computational spectrometer has good accuracy and efficiency. The STD-Net has successfully enabled computational spectrometry free from finding highly non-correlated filters which diminished the difficulty of filter design and fabrication. The STD-Net may provide a new method for the development of computational spectrometers.

## Figures and Tables

**Figure 1 sensors-24-01553-f001:**
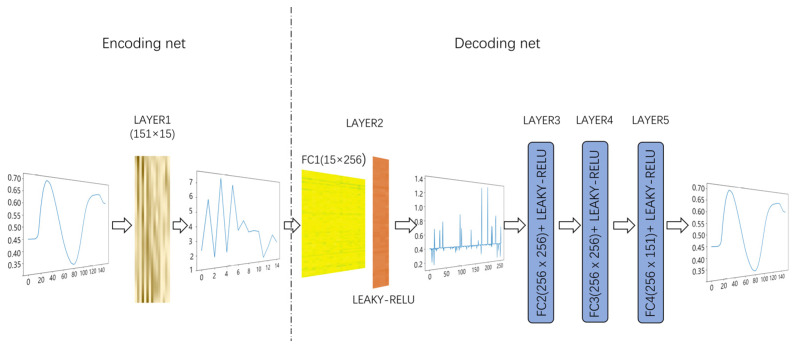
The architecture of STD-Net.

**Figure 2 sensors-24-01553-f002:**
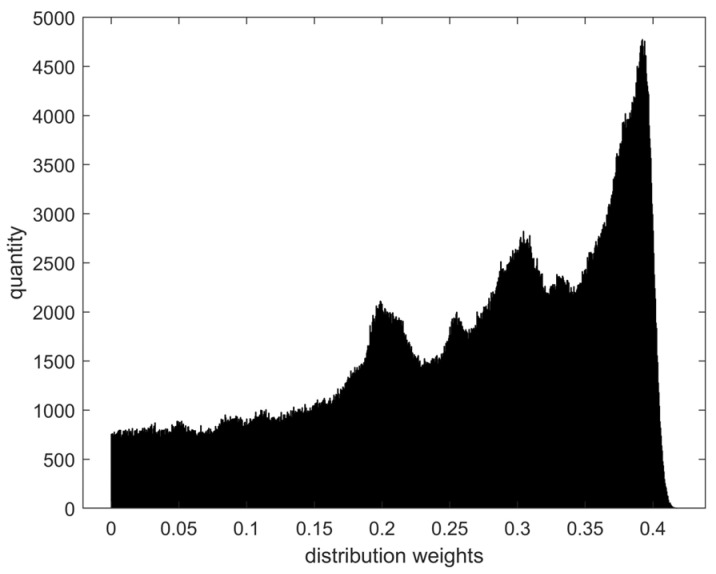
Spectral distribution of the entire dataset of CAVE and ICVL.

**Figure 3 sensors-24-01553-f003:**
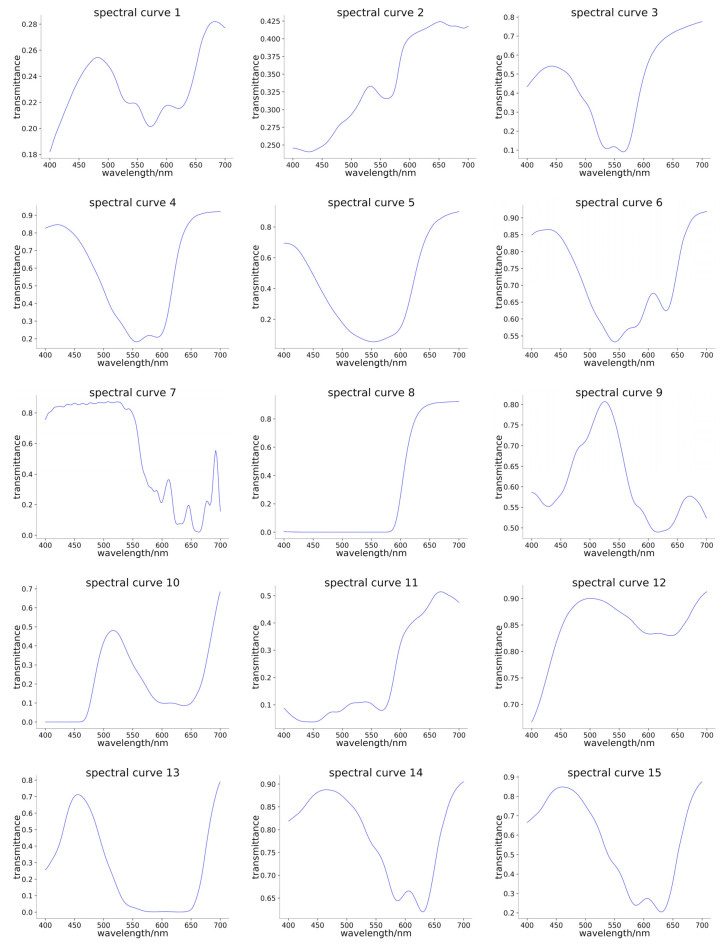
The fifteen selected optical filter transmittance curves.

**Figure 4 sensors-24-01553-f004:**
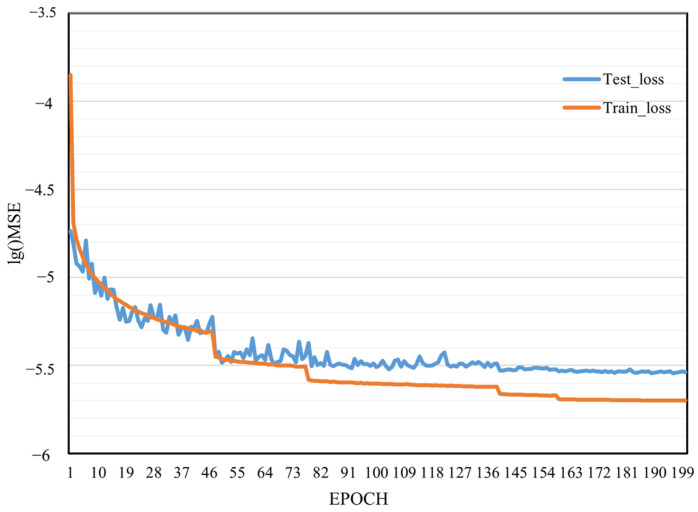
Training loss and testing loss.

**Figure 5 sensors-24-01553-f005:**
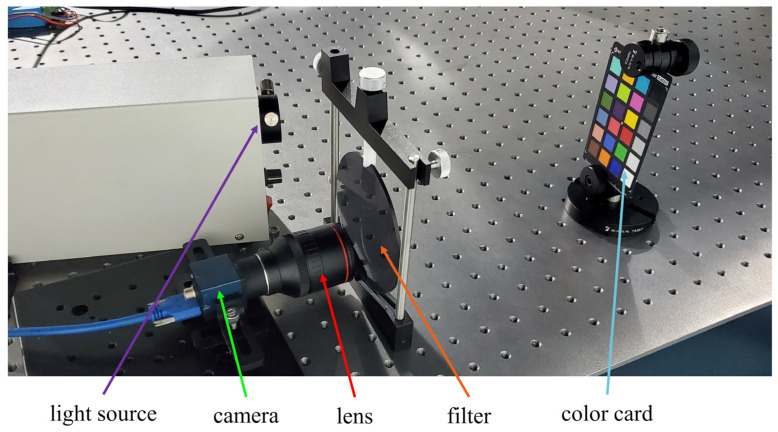
Experimental system.

**Figure 6 sensors-24-01553-f006:**
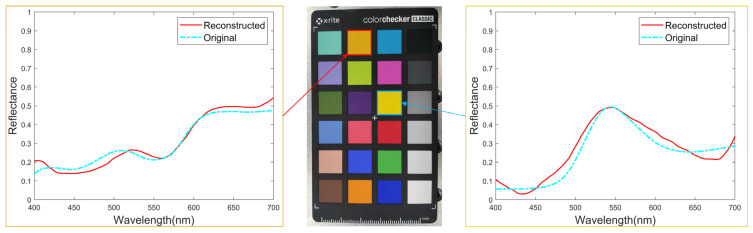
Results of spectral curves reconstruction.

**Table 1 sensors-24-01553-t001:** Four groups of the dataset.

	Training		Testing		Simulation	
	Proportion	Amount	Proportion	Amount	Proportion	Amount
Group-large	90%	1,450,000	9%	149,007	1%	15,993
Group-α	10%	164,540	89%	1,469,467		
Group-β	5%	82,021	94%	1,551,986		
Group-γ	3%	49,007	96%	1,585,000		

**Table 2 sensors-24-01553-t002:** Correlation coefficients of filter transmittance curves after group-α training.

	1	2	3	4	5	6	7	8	9	10	11	12	13	14	15
1	1.00														
2	0.34	1.00													
3	0.57	0.42	1.00												
4	0.51	0.00	0.84	1.00											
5	0.49	0.25	0.88	0.95	1.00										
6	0.49	0.12	0.76	0.90	0.86	1.00									
7	0.18	0.89	0.52	0.15	0.38	0.03	1.00								
8	0.53	0.84	0.78	0.51	0.69	0.34	0.82	1.00							
9	0.06	0.41	0.67	0.41	0.55	0.42	0.69	0.56	1.00						
10	0.44	0.30	0.22	0.24	0.16	0.24	0.05	0.14	0.55	1.00					
11	0.49	0.93	0.71	0.36	0.58	0.23	0.88	0.97	0.54	0.19	1.00				
12	0.61	0.32	0.16	0.31	0.33	0.34	0.03	0.12	0.46	0.70	0.16	1.00			
13	0.53	0.49	0.34	0.58	0.38	0.71	0.49	0.13	0.09	0.02	0.25	0.10	1.00		
14	0.54	0.55	0.12	0.47	0.29	0.60	0.63	0.24	0.44	0.25	0.33	0.15	0.88	1.00	
15	0.52	0.58	0.17	0.51	0.31	0.64	0.63	0.24	0.37	0.18	0.34	0.10	0.92	0.99	1.00

**Table 3 sensors-24-01553-t003:** Simulation results of 4 groups.

**Evaluation Index**	group−large			group−α		
	σ=0	σ=0.001	σ=0.01	σ=0	σ=0.001	σ=0.01
MSE	2.96 × 10^−6^	3.29 × 10^−6^	5.49 × 10^−6^	5.18 × 10^−6^	6.63 × 10^−6^	1.60 × 10^−5^
FWHM	0.4 nm	0.6 nm	1.2 nm	0.6 nm	0.8 nm	1.4 nm
Peak amplitude error	1.75 × 10^−3^	3.43 × 10^−3^	1.88 × 10^−2^	2.93 × 10^−3^	2.91 × 10^−3^	4.81 × 10^−3^
Peak wavelength position deviation	1.85 nm	3 nm	7.5 nm	2.14 nm	3.29 nm	5.14 nm
Reconstruction speed	31.01 μs	30.10 μs	30.39 μs	30.98 μs	30.62 μs	30.57 μs
Evaluation Index	group−β			group−γ		
	σ=0	σ=0.001	σ=0.01	σ=0	σ=0.001	σ=0.01
MSE	7.63 × 10^−6^	9.95 × 10^−6^	3.31 × 10^−5^	9.93 × 10^−6^	1.01 × 10^−5^	7.53 × 10^−5^
FWHM	0.8 nm	1.1 nm	1.5 nm	1.2 nm	1.3 nm	2.0 nm
Peak amplitude error	3.03 × 10^−3^	2.84 × 10^−3^	7.02 × 10^−3^	2.10 × 10^−3^	2.31 × 10^−3^	6.83 × 10^−3^
Peak wavelength position deviation	2.57 nm	3.86 nm	7.43 nm	5.07 nm	4.93 nm	8.43 nm
Reconstruction speed	30.38 μs	30.21 μs	30.84 μs	30.28 μs	30.45 μs	30.24 μs

**Table 4 sensors-24-01553-t004:** Experiment results of four groups.

Evaluation Index	Group−large	Group−α	Group−β	Group−γ
MSE	3.25 × 10^−6^	5.975 × 10^−6^	8.39 × 10^−6^	1.83 × 10^−5^
FWHM	0.5 nm	0.7 nm	0.9 nm	1.4 nm
Peak amplitude error	2.58 × 10^−3^	3.62 × 10^−3^	4.68 × 10^−3^	6.12 × 10^−3^
Peak wavelength position deviation	1.97 nm	3.35 nm	3.61 nm	5.93 nm
Reconstruction speed	32.45 μs	30.73 μs	30.94 μs	30.48 μs

**Table 5 sensors-24-01553-t005:** Comparison of reconstruction algorithms.

	GPSR (with Gaussian Matrix)	GPSR (with Filter Matrix of STD-Net)	OMP (with Gaussian Matrix)	OMP (with Filter Matrix of STD-Net)	PCSED
MSE	1.95 × 10^−2^	9.116 × 10^−3^	3.54 × 10^−3^	6.46 × 10^−2^	5.413 × 10^−4^
Reconstruction speed	1.67 ms	3.42 ms	224.37 μs	243.97 μs	281.9 μs
	WER-Net	STD-Net (group-large)	STD-Net (group-α)	STD-Net (group-β)	STD-Net (group-γ)
MSE	9.374 × 10^−5^	3.85 × 10^−6^	4.369 × 10^−6^	4.891 × 10^−6^	5.029 × 10^−6^
Reconstruction speed	71.4 μs	30.33 μs	30.22 μs	29.87 μs	30.29 μs

## Data Availability

Data are contained within the article.

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
