# Peer review of "A Neural Network Computational Spectrometer Trained by a Small Dataset with High-Correlation Optical Filters"

_sensors, 2024, doi:10.3390/s24051553_

Round 1
Reviewer 1 Report
Comments and Suggestions for Authors
I read the manuscript with great attention. Several typing errors are present and all cross-references to figures are skipped.
The manuscript describes many previous works on the subject without clearly explaining what is new in this manuscript. The description of the model is somehow unclear and the whole manuscript is very difficult to be followed.
I would suggest a deep revision of the manuscript before publication
Comments on the Quality of English LanguageSeveral typing errors are present and all cross-references to figures are skipped.
Author Response
Point 1. I read the manuscript with great attention. Several typing errors are present and all cross-references to figures are skipped.
Response: On behalf of all authors, we appreciate your devoted attention to the details of the manuscript, typographical errors in the manuscript have been fixed, as well as errors in the citation of figures.
Point 2. The manuscript describes many previous works on the subject without clearly explaining what is new in this manuscript. The description of the model is somehow unclear and the whole manuscript is very difficult to be followed. I would suggest a deep revision of the manuscript before publication
Response: This study proposes an NN computational spectrometer with high-correlated optical filters. It consists of high-correlation optical filters for encoding and a neural network called STD-Net for decoding, which is trained on a small training dataset. First, we propose a presumption that the spectral has a specific distribution law, in which a small training dataset, composed of randomly extracted data from the entire dataset, can cover the main features. In addition, a tailored loss function and adaptive learning rate mechanism are introduced to improve the training efficiency and reconstruction accuracy. In the reconstruction network, we proposed a four-layer FC architecture network. Multi-scale training datasets are established to train the constructed neural network. Furthermore, we conducted four groups of comparison experiments with different data extraction ratios, indicating that STD-Net could achieve high reconstruction accuracy and robustness, even when the training data quantity was immensely limited. Finally, an experimental system was introduced and the results indicate that the proposed NN computational spectrometer has good accuracy and efficiency. The STD-Net has successfully enabled computational spectrometry free from finding highly non-correlated filters which diminished the difficulty of filter design and fabrication. The STD-Net may provide a new method for the development of computational spectrometers.
Reviewer 2 Report
Comments and Suggestions for Authors
The manuscript presents an innovative and well-structured study on computational spectrometry, introducing a novel approach using high-correlation optical filters and a small training dataset with the STD-Net neural network. The work effectively addresses the challenges of conventional spectrometers by leveraging compressed sensing principles. The theoretical foundation is solid, providing a clear motivation for the proposed methodology. The neural network architecture, training methods, and loss function choices are well-explained, demonstrating a thoughtful design. The dataset analysis and distribution, along with detailed simulation and experimental results, showcase the robustness and efficacy of the proposed approach. The comparison with other algorithms strengthens the argument for the superiority of STD-Net in terms of both reconstruction accuracy and efficiency. Practical implications and potential applications are discussed, contributing to the relevance of the work. Overall, the manuscript is technically sound, well-organized, and presents a valuable contribution to the field of computational spectrometry. The manuscript can be considered for publication in Sensors with necessary revisions.
1. It is better to avoid using an abbreviation in the title. In addition, as the abbreviation “Neural networks (NNs)” has been defined in Line 60, the “(NN)” at the end of Line 61 can be removed.
2. Please improve the presentation of figures and tables. For Figure 1, the coordinate axes of the corresponding curves on the insets cannot be clearly seen. The title of Table 2 can be moved to the next page.
3. Could the authors further clarify how the distribution weights are evaluated and how they contribute to the overall distribution pattern to facilitate the readers’ understanding?
4. Could the authors provide additional details on how the average reconstruction accuracy MSE is used in the learning rate adaptive decay mechanism and offer insights into the decision-making process for adjusting the learning rate based on MSE?
5. In the discussion about the MSE plateau at 3E-6 during the middle stage of training, please provide additional clarification on its significance. Why is this level considered sufficiently high for reconstruction accuracy, and how does it impact the overall training process?
Comments on the Quality of English LanguageMinor editing of English language required
Author Response
Point 1. It is better to avoid using an abbreviation in the title. In addition, as the abbreviation “Neural networks (NNs)” has been defined in Line 60, the “(NN)” at the end of Line 61 can be removed.
Response: The abbreviation "NN" in the title has been changed to "Neural Network" and "(NN)" at the end of line 61 has been deleted.
Point 2. Please improve the presentation of figures and tables. For Figure 1, the coordinate axes of the corresponding curves on the insets cannot be clearly seen. The title of Table 2 can be moved to the next page.
Response: Figure 1 has been changed to a version with clear axes, and the title of table 2 has been moved to the next page.
Point 3. Could the authors further clarify how the distribution weights are evaluated and how they contribute to the overall distribution pattern to facilitate the readers’ understanding?
Response: The description of the distributional weights has been updated in Section 3.1 to "The correlation coefficients between each spectral curve and the other spectral curves are taken as the absolute average value and are considered as the distribution weight of that spectral curve in the large training dataset to derive the contribution of that spectral curve to the richness of the whole dataset. The lower the distribution weight, the higher the contribution."
Point 4. Could the authors provide additional details on how the average reconstruction accuracy MSE is used in the learning rate adaptive decay mechanism and offer insights into the decision-making process for adjusting the learning rate based on MSE?
Response: In the learning rate adaptive decay mechanism, the learning rate is reduced by 50% when the average reconstruction accuracy MSE does not decrease in two consecutive epochs.
Point 5. In the discussion about the MSE plateau at 3E-6 during the middle stage of training, please provide additional clarification on its significance. Why is this level considered sufficiently high for reconstruction accuracy, and how does it impact the overall training process?
Response: We have rewritten it as “In the middle of training, the downward trend of the loss function of the test has been reduced to close to 0. Subsequent training is less helpful in reducing the loss function of the test, which can be assumed that the test error at this time is the same as the training error at the end of training. At this time, the MSE of the test reaches 3E-6, and the comparison in Table 3 can be found to be a sufficiently high reconstruction accuracy” and highlighted it.
Reviewer 3 Report
Comments and Suggestions for Authors
This paper is an interesting study of computational spectroscopy and specifically the application of a neural network architecture for the estimation of spectral information from spectrometers incorporating high correlation optical filters. The primary assertion in the paper is that reliable spectral reconstruction can be achieved with relatively small training data sets when high correlation optical filters are employed.
The deployed neural network architecture, which the authors designate the description ‘STD-Net’, is not novel and has been used elsewhere where there are severe constraints on the extent of the training dataset. The motivation and potential applications of this work is well articulated by the authors and there clearly are advantages to a computationally efficient computational spectrometry architecture for a variety of emerging applications.
The hypothesis is that the NN architecture adopted allows robust reproduction of spectral information for real-world applications and the core assumption is that the available spectral data sets (here CAVE and ICVL) are sufficiently representative to assess the generic efficacy of the approach.
The simulation and testing methodologies outlined in the paper are reasonable, and relevant metrics including accuracy and robustness are fairly assessed for highly constrained data sets. Whilst the paper is interesting and useful there remain several unanswered questions, including the reliability of assessing efficacy based on CAVE and ICVL. The claims made by the authors are not exaggerated and there is an acknowledgement that this work is exploratory, however the paper would benefit from a more convincing discussion on the overall validity and limitations of the approach for general real-world applications.
Comments on the Quality of English LanguageThe English grammar and language is of varying quality throughout the paper and needs another edit.
Author Response
Point 1. The simulation and testing methodologies outlined in the paper are reasonable, and relevant metrics including accuracy and robustness are fairly assessed for highly constrained data sets. Whilst the paper is interesting and useful there remain several unanswered questions, including the reliability of assessing efficacy based on CAVE and ICVL. The claims made by the authors are not exaggerated and there is an acknowledgement that this work is exploratory, however the paper would benefit from a more convincing discussion on the overall validity and limitations of the approach for general real-world applications.
Response: The CAVE dataset covers the spectra of five kinds of scenes: objects, skin, hair, paint, food, and drink, and has been cited by more than 632 pieces of literature. ICVL dataset covers the spectral information of nature, trees, and buildings, and has been cited by more than 415 pieces of literature. The mixture of the two, with a total of 1.65 million spectral curves, contains spectral information of most commonly used scenes and is also widely recognized and utilized by the image processing industry
Round 2
Reviewer 1 Report
Comments and Suggestions for Authors
The authors have efficiently addressed the unclear points. The manuscript is now acceptable for publications on Sensors